# Outdoor Air Pollution and Childhood Respiratory Disease: The Role of Oxidative Stress

**DOI:** 10.3390/ijms24054345

**Published:** 2023-02-22

**Authors:** Arianna Dondi, Claudio Carbone, Elisa Manieri, Daniele Zama, Chiara Del Bono, Ludovica Betti, Carlotta Biagi, Marcello Lanari

**Affiliations:** 1Pediatric Emergency Unit, IRCCS Azienda Ospedaliero-Universitaria di Bologna, Via Massarenti 9, 40138 Bologna, Italy; 2Italian National Agency for New Technologies, Energy and Sustainable Economic Development (ENEA), 40129 Bologna, Italy; 3Specialty School of Pediatrics, Alma Mater Studiorum, University of Bologna, 40126 Bologna, Italy; 4Department of Medical and Surgical Sciences (DIMEC), University of Bologna, 40138 Bologna, Italy

**Keywords:** children, air pollution, air quality, oxidative stress, respiratory disease, lung function

## Abstract

The leading mechanisms through which air pollutants exert their damaging effects are the promotion of oxidative stress, the induction of an inflammatory response, and the deregulation of the immune system by reducing its ability to limit infectious agents’ spreading. This influence starts in the prenatal age and continues during childhood, the most susceptible period of life, due to a lower efficiency of oxidative damage detoxification, a higher metabolic and breathing rate, and enhanced oxygen consumption per unit of body mass. Air pollution is involved in acute disorders like asthma exacerbations and upper and lower respiratory infections, including bronchiolitis, tuberculosis, and pneumoniae. Pollutants can also contribute to the onset of chronic asthma, and they can lead to a deficit in lung function and growth, long-term respiratory damage, and eventually chronic respiratory illness. Air pollution abatement policies, applied in the last decades, are contributing to mitigating air quality issues, but more efforts should be encouraged to improve acute childhood respiratory disease with possible positive long-term effects on lung function. This narrative review aims to summarize the most recent studies on the links between air pollution and childhood respiratory illness.

## 1. Introduction

Outdoor air pollution represents one of the leading environmental risks to human health. Exposure to air pollutants is ubiquitous, and according to the World Health Organization (WHO), every day, around 93% of children under the age of 15 years breathe polluted air, which puts their health at serious risk [1]. Even low levels of air pollution harm children’s lung function and growth and increase the incidence of respiratory diseases such as asthma, bronchiolitis, respiratory infections, and bronchitis. 

Air pollution involves chemical and biological materials discharged into the atmosphere [2] and is characterized by a complex mixture of gases and particles whose sources and composition are strictly related [3] and may vary substantially spatially and temporally. Particulate matter (PM) with an aerodynamic diameter of 2.5 µm (PM_2.5_) or 10 µm (PM_10_), ozone (O_3_), nitrogen oxides (NO, NO_2_), sulfur oxides (SO, SO_2_), and carbon monoxide (CO) are the principal conventional metrics used to quantify exposure to air pollution. Computing air pollution levels provide a measure of exposure used as a surrogate for risk. Numerous studies have linked such air quality data to adverse health outcomes. It is also used to advise vulnerable groups (such as children or the elderly) on risk management. It follows that the closer the metric is to the actual harmful component of the exposure, the better the risk management and the relationship to adverse health effects in epidemiological studies are likely to be. When exposure is transmitted into an internal dose, mechanistic toxicologists, whose job is to address biological mechanisms, identify the proper harmful entity in the dose as the biologically effective dose (BED). The BED is the entity that drives the adverse effect(s). The gap between the BED and the total dose, as derived from the exposure metric, can be considerable, especially in the case of ambient particles [4]. Among air pollutants, PM_2.5_ has received particular attention. There is evidence of causal relationships between exposure to PM_2.5_ air pollution and all causes of mortality and several diseases, including lung cancer, stroke, respiratory infections, and pulmonary diseases [5,6]. In the past 15 years, most epidemiological and toxicological studies have focused on the inhalation of airborne PM, which was found to have a stronger correlation with short-term (premature mortality, hospital admissions) and long-term (morbidity, lung cancer, cardiovascular and cardiopulmonary diseases, etc.) adverse health effects compared with other atmospheric gas pollutants. [7]. In most studies in recent decades, the mass of PM has been the metric chosen to demonstrate associations between ambient pollution and a wide range of outcomes (e.g., Pope and Dockery, 1999 [8]). However, current research on airborne particle-induced health effects investigates the critical characteristics of PM that determine their biological effects and agrees that PM_2.5_ (or PM_10_) mass is not ideal but represents a surrogate for the BED. This debate is based on the fact that much of the ambient particle mass consists of low-toxicity components, such as inorganic salts and mineral dust [9]. These contribute substantially to the mass metric but, except in rare circumstances, do not contribute substantially to the BED. In contrast, relatively tiny masses of transition metals, polycyclic aromatic hydrocarbons (PAHs), and other organic reactive species may significantly contribute to the BED [10,11,12] and determine the potential to elicit inflammatory injury, oxidative damage and other biological effects. 

Children are more vulnerable to the harmful effects of air pollution than adults because their lungs are still developing. Lung development in children is a long-term process that starts in utero and continues through life. The highest percentage of alveoli, respiratory bronchioles, epithelium, and immune cell populations develop in the post-neonatal period, and changes in the lung continue through adolescence. This suggests that during the first years of life, the lung is highly vulnerable to acute injury [12,13,14].

The pediatric population is also more vulnerable to air pollutants due to (1) a lower efficiency of oxidative damage detoxification systems, (2) a higher metabolic and breathing rate, and (3) enhanced oxygen consumption per unit of body mass. In addition, children spend more time outdoors than adults, so they have increased exposure to outdoor air pollution [13]. Furthermore, pollutant damage probably starts with prenatal exposure, as much research shows that exposure to pollutants during pregnancy can affect fetal development and babies’ long-term respiratory health [15].

In addition to studies highlighting the short-term effects of exposure to air pollution on lung function, many studies suggest long-term adverse effects on lung function and growth [16].

The purpose of this narrative review is to summarize the most recent research on air pollutants and their impact on childhood respiratory disease, with a focus on the major outdoor air pollutants and their composition, the pathogenic and biological mechanisms by which pollutants affect health, and the impact on children’s respiratory systems in the short and long term.

## 2. Main Outdoor Air Pollutants and Their Composition

### 2.1. Air Pollution and Health Effects

Numerous studies have documented associations between air pollution and mortality and morbidity in adults and children [17], with fine PM playing a major role. Nevertheless, fundamental uncertainty and disagreement persist regarding which physical and chemical properties of particles (or unidentified confounding environmental influences) have the most significant impact on health, the underlying pathophysiological mechanisms, and which specific air quality standards should be adopted to minimize risks for public health. This makes causal links between health effects caused by air pollution exposure and identifying proper properties to use as metrics challenging to establish within the scientific community. Moreover, air quality is regulated by most jurisdictions in terms of individual components and using single markers or integrated measurements (for instance, PM_10_ or PM_2.5_ daily mass), which are now recognized as not ideal for representing the multiple and specific factors involved in the pathogenesis of non-communicable diseases resulting from exposure to air pollution as a whole. The ambient atmosphere is a dynamic system in which the mixture of air pollutants changes quickly over time. The fluctuation of important atmospheric parameters influencing ambient PM concentrations and chemical composition (thus, human exposure), such as particle sources’ emission strengths, gaseous precursors, temperature, relative humidity, mixing height, wind direction, and speed, occurs on time scales much shorter than a few hours. For this reason, daily air quality monitoring cannot be adequate to fully assess the physicochemical properties of atmospheric pollution involved in the causal mechanisms associated with the BED. In the last 15 years, there has been a growing literature pointing to potential alternative metrics, and a particular focus has been dedicated to specific health effects in association with cellular oxidative stress initiated by the formation of reactive oxygen species (ROS) at the surface of and within target cells [4,10]. The main conventional air pollutants and their compartmental deposition are summarized in Figure 1, while their corresponding primary sources and effects on the respiratory system are listed in Figure 2. 

### 2.2. Gases

Highly reactive gaseous pollutants—such as O_3_, NO, NO_2_*,* SO, and SO_2_—are still being studied for their health effects because they are linked to a cascade of inflammatory processes. They are produced mainly by anthropogenic sources and are thus tracked and regulated by governments and air-quality monitoring agencies. Ground-level O_3_ is typically formed by photochemical reactions between solar radiation and other gaseous precursors, such as NO, NO_2_, and volatile organic compounds (VOCs), especially in warm summer conditions. NO, and NO_2_ are formed primarily by the reaction of O_3_ with nitric acid emitted during fossil fuel combustion. The primary sources of SO and SO_2_ in the developed world are primary emissions during energy production or industrial processes. Although exposure to SO_2_ has been greatly reduced in the developed world through treatment for emissions from coal-fired power plants and energy sources other than coal combustion, it remains an issue in developing countries (e.g., China). These gaseous pollutants vary substantially in their environmental persistence and oxidizing capacity; whereas O_3_ is a potent oxidizing agent, NO_2_ is a relatively weak oxidant. Controlled NO_2_ exposure appears to have a mild airway inflammatory effect at high levels of exposure that are unlikely to occur in non-experimental settings, but it causes minor lung inflammation at typical ambient concentrations. NO_2_ is supposed to have a deeper penetration into the respiratory tree, which could cause microlesions in the respiratory tract and increase susceptibility to infections [18]. However, much debate, the so-called “NO_2_ dilemma”, surrounds whether the epidemiological associations found for fossil combustion-related air pollution are due to direct effects of NO_2_ or if it is just a marker and the causes of pulmonary diseases are associated with the broader mixture of correlated components. Indeed, studies report strong associations between NOx and health outcomes (e.g., pediatric asthma) that mimic other specific properties of traffic PM [19], such as BC and particle number concentrations. Because SO_2_ is a reductant, it probably causes pulmonary diseases through a different mechanism. Extensive research has been conducted on the effects of short-term controlled exposure to O3 and SO2 at relevant concentrations [20,21]. O_3_ exposure results in airway inflammation, airway hyper-responsiveness, and lung function decrees in healthy and fragile individuals, whereas SO_2_ causes more prominent bronchoconstriction, mostly in fragile individuals [22].

### 2.3. Particulate Matter (PM)

PM is the crucial ingredient in polluted air concerning increases in cardiac and respiratory morbidity and mortality. To prevent this staggering loss of life and cost to society, it is essential to understand the characteristics of the toxic particles and gain insight into how they are related to adverse health effects. PM is composed of solid and liquid particles, which can be classified into coarse, fine, or ultrafine (UFPs or PM_0.1_) particles based on their size and mean diameters: coarse particles have a diameter of more than 2.5 μm, fine particles are considered those with diameters in the range 0.1–2.5 μm, and ultrafine particles have diameters <0.1 μm. Particles are characterized by a myriad of primary and secondary components and complex chemical mechanisms, encompassing both natural and anthropogenic sources. These lead to very heterogeneous physical and chemical properties, varying in relation to the mix of pollutants/components, season, and geography.

It is currently possible to measure many properties of PM populations in the atmosphere. Size-specific mass, surface area, the total number of particles, and the number of particles in different size ranges are all currently measurable, as is the chemical composition of particles in the atmosphere. All these properties, which vary in space and time, can help to describe the multiple characteristics of particle populations and are reported to play a role in the adverse health effects of air pollution, including allergy, asthma, cardiovascular, and respiratory diseases. However, the functional mechanisms explaining the causative relations and mechanisms of interaction on the molecular level remain mostly unclear. There is no conclusive evidence to pinpoint a single pollutant or a limited number of species as the main harmful components/properties of PM, and further research in this area is ongoing to inform policy priorities [23].

Currently, government and air-quality monitoring agencies track and regulate the size-specific mass of 10-μm-diameter (PM_10_) and 2.5-μm-diameter (PM_2.5_) particles as specified by the Code of Federal Regulations in the US and the European Committee for Standardization (CEN) standard in Europe. As these or equivalent methods are used in routine air quality monitoring networks, they have formed the basis of numerous epidemiological studies. However, the WHO report (2021) [5] review provides an authoritative starting point for considering possible future metrics for regulating particles in the ambient atmosphere. Another mass metric, PM_1_, has been suggested as possibly valuable for managing PM levels in the atmosphere, mainly because it provides better separation of the coarse mode and accumulation mode (and ultrafine) fractions than the 2.5 micron cut-off. Furthermore, WHO prioritized specific types of PM, i.e., Black Carbon (BC), Elemental Carbon (EC), heavy metals, and the concentration of ultrafine particles, but concluded that the quantitative evidence of independent adverse health effects from these pollutants was still insufficient for new air quality guideline levels [5].

In 2021, the WHO [5] recommended lowering the PM_2.5_ annual air quality guideline level from 10 to 5 µg m^3^ to reflect the new evidence about effects occurring at low levels of exposure. Among the reasons possibly explaining the occurrence of health effects even at these very low doses is that these studies are based on PM_2.5_ mass, a metric not ideal for representing the BED of toxic PM_2.5_ [24]. In fact, despite the PM mass being the same, health impacts can vary significantly depending on its chemical composition and size distribution. For instance, the content of relatively small masses of transition metals and organic species increases their intrinsic toxicity, as does the abundance of ultrafine particles (even if they represent a minor contribution to their total mass). These findings led the scientific community to investigate more health-relevant aerosol metrics than PM mass, looking at the link between physicochemical features of the atmosphere and specific relevant health outcomes.

An important new area of research has emerged in the past 15 years or so, undertaken in several different countries, using a range of different approaches, which has demonstrated that the mechanism leading to oxidative stress can be identified as a unifying feature underlying the toxic actions linking PM chemical composition to biological effects [25,26]. Oxidative stress is based on a homeostatic mechanism implying protective responses and can be defined as a change to living cells (and thereby the organs and tissues composed of those cells) caused by reactive oxygen (or nitrogen) species. It is also defined as an imbalance between reactive oxidant species (ROS) generated from the stimulation of pro-inflammatory factors and antioxidants, such as glutathione and antioxidant enzymes, with the former prevailing. Thus, the oxidative potential is recognized as an essential property of PM and has gained growing attention; several now widely used assays and techniques for quantifying particle ROS activity have been developed, and ROS studies have received particular attention. In addition, a wide range of experimental conditions has been explored in the laboratory to assess PM characteristics in terms of toxicological endpoints and mechanisms of ROS formation [27].

Numerous studies have shown a complex relationship between PM oxidative and pro-inflammatory properties and primary combustion aerosol concentrations. They indicate that these are significantly impacted by changes in the particles’ size, mixing, and aging state in the real atmosphere. The best candidates for particulate compounds responsible for ROS activity encompass transition metals and specific carbonaceous compounds, such as quinones species and BC [28] or some combination of different physicochemical properties associated with specific PM sources and atmospheric conditions.

For example, during the fall and winter seasons, when common respiratory disease epidemics occur, PM accumulates in many densely populated regions of continental Europe and, more broadly, in many mid-latitude areas, concomitance with the development of high-pressure atmospheric structures. These conditions favor the condensation of semi-volatile species, causing high PM episodes mainly accounted for by carbonaceous matter and ammonium nitrate [29]. Biomass burning for residential heating and fossil fuel combustion dominate as sources of primary organic PM in urban environments, leading to higher risks for detrimental health effects [30]. A higher BC to total organic PM ratio and a higher concentration of ultrafine particles can characterize this PM and are valuable additional air quality indicators to evaluate the health risks of air quality dominated by primary combustion particles [19,31]. A few studies have also highlighted the toxicological potential of secondary organic PM (SOA), demonstrating that ambient PM is responsible for potential impacts on human health and is not only emitted from pollution hotspots [32]. Other authors, comparing ROS assays performed on size-segregated PM samples from six cities on three continents, showed that finer aerosol size fractions tend to have a higher ROS activity and that chemical components determining ROS formation include several transition metals (primarily emitted) and polar organic compounds (likely of secondary origin) [33].

During the CARE experiment carried out in the urban area of Rome (Italy) in wintertime [10], the most significant pro-inflammatory and oxidative responses have been related to a specific ultrafine particle type enriched in BC and associated with new emissions of fossil fuel combustion. Different PM features, characterized by the same levels of BC content but a different mixing of fossil fuel ultrafine particles with background PM, exhibited low oxidative responses, even with higher PM_2.5_ mass concentrations. As exemplified in Figure 3, the mixing state is considered one of the most critical factors determining the impacts of particles not only on the cloud–climate interactions but also on human health [34]. Furthermore, mixing structures of different species or types of change depend on the aerosol source, pollution level, transport distance, and relative humidity, and these changes tend to be more rapid in a more polluted environment [34]. This suggests the importance of considering specific PM features in terms of chemical composition, mixing state, and size distribution associated with specific sources instead of conventional metrics (e.g., PM mass or single pollutant concentrations) that can obscure relevant confounding factors, co-emitted pollutants, and environmental variables, as well as the related health effects of the atmospheric mixture as a whole [35,36].

In conclusion, recent research indicates that the ROS generated by atmospheric processing significantly impacts the PM oxidative potential, which appears to play an important role in aerosol health effects and provides a direct link between atmospheric and physiological multiphase processes. Consequently, different techniques for its quantification are being tested as an alternative air quality metric in a significant number of studies worldwide. However, it remains the subject of intense research and discussion [37].

### 2.4. Traffic-Related Air Pollution (TRAP)

Road traffic is a significant source of air pollution in urban environments. In recent decades, many studies linking air pollution and health effects in the urban atmosphere have focused on traffic-related air pollution (TRAP). Figure 4 exhibits the typical patterns of TRAP exposure across an urban environment [38], suggesting how the different contributors of pollution at different scales (e.g., urban primary, urban, regional, continental background, or long-range transport sources) may affect TRAP physicochemical properties and the associated toxicity. TRAP is produced by the combustion of fossil fuels and is composed of a complex mixture of pollutants (NO, NO_2_) and particles containing both non-volatile and semi-volatile components. The semi-volatile compounds may partition between the gas and particle phases in ambient conditions depending on their vapor pressure and degree of atmospheric dilution and participate in photochemical reactions in the atmosphere [39].

Consequently, the physical and chemical characteristics of combustion-generated PM change dramatically with background conditions, atmospheric dilution, and processing, which may alter their toxic properties and influence their role in population exposures and public health. Combustion particles also derive from various sources other than motorized road traffic, including wood and coal burning, shipping, and industrial sources, and these sources may contribute significantly to ambient combustion particle concentrations, even in urban areas. As mentioned in the previous section, semi-volatile PM contains a wide variety of organic micropollutants, many of which possess genotoxic and carcinogenic characteristics, such as dioxins, PAHs, and their derivatives. Some of these semi-volatile organic species trigger a chain of biochemical reactions in cells, change their redox state, and exert oxidative stress. Polar organic compounds, such as quinones, oxygenated PAHs, and aldehydes, are reported to induce oxidative stress in cells. BC, also known as soot, is a significant source of combustion PM and has been repeatedly proposed as a PM metric candidate to reflect better the health effects of combustion-related air pollution than PM mass because it is relatively easy and not expensive to measure [31]. Traffic-related PM is characterized by fresh UFPs enriched in BC and micropollutants, making this PM more relevant in terms of PM toxicity per unit mass because of its higher oxidative potential and smaller size. This indicates the potential for particles with high surface reactivity and adsorbed toxic molecules to deposit in the deepest tracts of the respiratory system (more easily with decreasing size) [40]. Additionally, non-tailpipe emissions (pollutants emitted from evaporative emissions of fuel, resuspension of dust, wear of brakes and tires, and abrasion of road surfaces) are reported to exert some relevant effects on health [40].

TRAP is associated, with children, with low birth weight, Intrauterine Growth Restriction (IUGR), preterm birth, asthma exacerbation, chronic asthma, and lower respiratory infections [41]. Moreover, TRAP exposure in pediatric patients is associated with metabolic syndrome (glucose deregulation, blood pressure), oxidative stress [24], and atherosclerosis [42].

## 3. Health Outcomes: Effects on Childhood Respiratory Health

Air pollution can cause respiratory diseases and affect lung function and growth, depending on the type of pollutant, its concentration, and its dimensions [3]. Pollutants can harm children’s respiratory health through various mechanisms, including local and systemic inflammation, oxidative stress damage, immune response modulation, and genetic changes [2,43]. Children are particularly susceptible to the adverse effects induced by air pollution because (1) their respiratory and immune systems are not fully developed [44]; (2) their airway epithelium is more permeable to pollutants [45]; (3) they have higher resting metabolic rate, higher respiratory rate and a higher rate of oxygen consumption because they are growing and because they have a larger surface area per unit body weight; (4) they also have a smaller lung surface area per kilogram, so breathing air reaches a relatively smaller area of the lung; (5) they have smaller airways, so pollutants’ irritation can cause a potentially remarkable obstruction; finally, (6) obese children have a higher respiratory rate that causes a higher deposition of fine particles in the lungs; consequently, children with higher body mass indexes might be more vulnerable to air pollution damage [45].

To study the impact of acute exposure to pollutants on the developing lung, Fanucchi and colleagues examined the postnatal lung morphogenesis and function of a group of infant monkeys—which have human-like airway structure and postnatal lung development—subjected to cyclic exposure to O_3_. Compared with controls, O_3_-exposed monkeys had markedly increased airway resistance at baseline, fewer airway generations, hyperplastic bronchiolar epithelium, and impaired smooth muscle in the terminal and respiratory bronchioles. These results suggest that episodic exposure to environmental pollutants alters postnatal tracheobronchial airway morphogenesis and development [46].

The main direct and indirect damage mechanisms to the respiratory system are represented in Figure 1 and Figure 5.

### 3.1. Local and Systemic Inflammation and Oxidative Stress Damage

Air pollutants trigger pulmonary oxidative stress and inflammation pathways [47]. The defense of the respiratory system against inhaled toxicants consists of innate mechanisms such as aerodynamic filtration, mucociliary clearance, particle transport, and detoxification by alveolar macrophages, as well as innate and acquired systemic antiviral immunity [48]. In experimental models, some of these functions, such as pulmonary inflammation, chemokine expression, and airway hyper-responsiveness, appear to be modified by exposure to pollutants. For example, phagocytosis of pollutant particles by alveolar macrophages produces inflammatory cytokines and chemokines, which are then released into the systemic circulation [49], causing local and systemic inflammation.

The pro-inflammatory state promoted by pollutants is primarily due to the oxidative stress resulting from the production of ROS and the reduction of endogenous antioxidants. Oxidative stress can directly damage cellular proteins and DNA, create neoantigens by protein oxidation, and activate pro-inflammatory signaling cascades, such as NF-kB and MAPK pathways. Oxidative stress affects Antigen Presenting Cells (APCs) function and co-stimulation, T helper (Th) lymphocyte imbalance favoring Th2 response, and interferon (IFN)-gamma production [50]. Alveolar macrophages react to exposure by producing ROS, nitrogen species and releasing tumor necrosis factor (TNF)-α and interleukin (IL)-1, which contribute to an imbalance between oxidants and antioxidants [3]. Then free radicals attack local tissue components, causing cellular damage [26].

In response to the inflammatory state, children chronically exposed to high levels of air pollutants show structural changes in the airway mucosa. A study by Calderón-Garcidueñas and colleagues compared by electron microscopy the nasal epithelium of children living in a polluted city with that of peers living in a city with low levels of air pollution [51]. The first group’s nasal epithelium revealed metaplastic cells, ciliary dyskinesia, epithelial junction deficiencies, and PM deposited in intercellular spaces. These modifications might result in a lower capacity to protect the lower respiratory tract and make the distal airways more vulnerable to pollutants. O_3_, PM, and aldehydes were identified as the most responsible pollutants [51].

Long-term exposure to pollutants leads to a loss of lung function due to progressive airway remodeling and fibrosis of the lung due to oxidative stress, according to several animal model studies [49,52]. Moreover, oxidative stress damage seems to be related to the primary development of asthma and chronic obstructive pulmonary disease (COPD) [47].

### 3.2. Immune Response Modulation

Cell studies and animal models proved that air pollution could affect both innate and adaptive immune systems (Table 1) because the respiratory tract represents the first interface between the immune system and the environment, reducing their ability to limit the spread of infectious agents.

Air pollutants damage epithelial cells, induce granulocyte and mast cell migration and activation, enhance pro-inflammatory cytokines (IL-1, IL-6, and TNF) production, and reduce macrophage phagocytosis and clearance. Pollutants stimulate a pro-inflammatory immune response by triggering several pathways, including Toll-Like Receptors (TLRs), which detect pathogen-associated molecular patterns of infectious microorganisms and analogous injurious triggers, like pollutants. PM, for example, can activate the TLRs pathway because they frequently transport microbial molecules such as lipopolysaccharide and fungal spores, as well as cellular damage products like oxidized phospholipids and nucleic acids [50].

Pollutants enhance the Th2 immune response, IgE, IL-5, IL-13, and leukotriene production, resulting in asthma and allergies exacerbation and an inappropriate antimicrobic environment. Air pollution can also impair anti-viral immune responses by suppressing the Th1 pathway, deregulating IFN-gamma, and reducing macrophage phagocytosis and clearance capacity. For instance, NO_2_ increases Th2 cytokines and ICAM-1 (Intercellular Adhesion Molecule-1) epithelium expression, facilitating respiratory infections since ICAM-1 is a receptor for Rhinovirus and respiratory syncytial virus (RSV) [50].

Pollutants can reduce immune tolerance because pro-inflammatory cytokines reduce regulatory T cell responses. Protein oxidation leads to neo-antigen formation, triggering autoinflammatory and autoimmune diseases, such as systemic lupus erythematosus, rheumatoid arthritis, multiple sclerosis, and type 1 diabetes mellitus. PM is proven to act as an adjuvant on APCs, increasing the immunogenicity of antigens [50].

### 3.3. Genetic and Epigenetic Changes

Epigenetic changes regulate the expression of many genes without changing the DNA sequence itself, including those involved in the inflammatory immune response and lung growth. Environmental factors can alter epigenetic states, and a growing number of studies are looking into the effects of air pollutants on epigenetic mechanisms. DNA methylation, which involves the attachment of a methyl group to the C5 position of cytosine to form 5-methylcytosine, is a crucial epigenetic mechanism that modulates gene expression [53] and seems to act as a mediator between air pollutants and their effects on the respiratory system [54].

The effects of air pollution on DNA methylation have been investigated both in the prenatal and postnatal periods. Several studies analyzed samples obtained from the placenta or cord blood and showed that environmental exposure to air pollutants during pregnancy has a significant impact on DNA methylation in newborns [55], with particular attention to the effects of NO_2_ [56], PM_2.5_ [57], and PM_10_ [58]. In addition, they report differences related to the type of pollutant and the stage of development upon exposure, e.g., PM has been shown to have a more significant impact in the initial period of gestation [58,59]; NO_2_ exposure was associated with DNA methylation changes in mitochondria-related genes and antioxidant defense pathways [56].

A systematic review conducted by Isaevska and colleagues showed that prenatal exposure to different pollutants (PM, NO_2_, O_3_, CO, SO_2_, VOC, BC, EC, OC, and PAH) is associated with global methylation loss, telomere reduction, and epigenetic alterations involving genes related to oxidative stress response, mitochondrial function, inflammation, and fetal growth [60].

Exposure to air pollutants may also lead to epigenetic alterations of genes related to the inflammatory pathway [61]. For example, a study on mice that combined inhaled diesel exhaust particles and allergen exposure found changes in the methylation of Th2 cytokines and IgE production in vivo, implying an underlying mechanism that may contribute to the development of asthma [62]. For instance, in infants with severe bronchiolitis, microRNA post-transcriptional regulation could explain the connection between pollution and asthma [63]. Furthermore, DNA methylation plays an essential role in TRAP-induced asthma [64].

## 4. Focus on Respiratory Diseases

Children’s respiratory health is the result of the interaction between environmental influences and genetic susceptibility factors [65]. The respiratory system is the principal interface between air and the organism, so it is a primary target for toxicants such as air pollutants [66]. In recent years, several outdoor air pollutants have been consistently linked to developing and exacerbating pediatric respiratory disorders such as asthma, bronchiolitis, and respiratory infections.

The effects of short- and long-term exposure to air pollution on respiratory diseases are summarized in Table 2.

### 4.1. Asthma

Asthma is the most common chronic respiratory disease in children, and its development seems to result from the interaction between genetic predisposition and environmental factors [96]. Childhood exposure to outdoor air pollution may play an essential role in exacerbating and the onset of chronic asthma [22,97]. A systematic review and meta-analysis by Khreis et al. [84] highlighted the connection between exposure to TRAP and the development of asthma in children. Another systematic review and meta-analysis by Bowatte and colleagues [78] found a link between TRAP exposure and an increased risk of sensitization to common allergens in pediatric patients. An association has been demonstrated between SO_2_, NO_2_, and PM_2.5_ and a significant decrease in small airway function and an increase in airway oxidative stress in asthmatic children [98].

NO_2_ is a significant trigger for pollution-induced asthma exacerbations, mainly because it increases leukotriene production [50]. In addition, several observational and epidemiological studies have shown that air pollutants, mainly O_3_, SO_2_, NO_2_, and UFPs [74], are associated with acute asthma exacerbations [99], and the risk is higher in children already affected by chronic asthma [75,76,77,100]. Likewise, a meta-analysis showed that exposure to ambient pollutants (especially PM_2.5_) increases the risk of asthma onset in healthy young children [78].

Furthermore, some studies have shown that the anti-pollution measures adopted during the SARS-CoV-2 pandemic and restrictions on the circulation of motor vehicles during public events can lead to a reduction in the primary environmental pollutants and a consequent decrease in emergency department visits due to asthmatic exacerbations [101,102].

### 4.2. Bronchiolitis

Bronchiolitis is the most common cause of lower respiratory infection and hospitalization in children younger than one year, and it is characterized by airway inflammation and obstruction of the lower respiratory tract [103,104]. Several studies have investigated the possible relationship between exposure to air pollutants and the incidence and severity of bronchiolitis due to the pollutants’ effects on the immune system and viral life cycles, especially that of RSV. A study conducted on a population of 18,595 infants highlighted that sub-chronic (the month preceding hospitalization) and chronic (lifetime) exposures to PM_2.5_ are significantly associated with an increased risk of hospitalization for bronchiolitis and that the risk depends on the concentration of PM (hospitalization risk nearly increased to 9% for each 10 μg/m^3^ increase in PM_2.5_) [105]. Another study discovered a link between the risk of hospitalization for RSV bronchiolitis and PM_10_ exposure, particularly in the two previous weeks [72], because this pollutant can be associated with viral particles and activate endocytic pathways, facilitating viral entry [69]. PM can also increase viral survival and inflammation, inducing IL-6 and IL-8 synthesis [69]. UFPs can increase viral replication and inflammation, enhancing nerve growth factor and chemokine production [69]. Likewise, a study conducted in Hong Kong on 29,688 infants referred to the hospital for acute bronchiolitis showed a link between an increased risk of hospitalization, NO_2_ and PM_10_ exposure, and high temperature, highlighting the combined effect of meteorological factors and air pollutants [18]. PM_2.5_ and PM_10_ levels are significantly associated with the severity of bronchiolitis [70]. RSV bronchiolitis incidence has a significant correlation with NO, NO_2_, PM_10_, and PM_2.5_ [68,71].

### 4.3. Respiratory Infections

Exposure to air pollution, especially PM, seems to be linked to upper and lower respiratory infections in children of any age [106,107], likely because polluted air deregulates anti-viral immune responses [50]. These effects disproportionately affect children with other respiratory disease, like asthma [108]. Upper respiratory infections are significantly associated with PM_2.5_, PM_10_, SO_2_, NO_2_, and CO [95], and childhood NO_2_ exposure has been linked to persistent cough [109]. A meta-analysis of ten European birth cohorts found relevant associations between air pollution exposure (NO_2_, OR 1.47; PM_10_, OR 1.77; PM_2.5_, OR 4.06) and pneumonia and identified the first year of life as the most vulnerable period [110]. Glick et al. demonstrated that greater O_3_ and PM_2.5_ were associated with more severe pneumonia [83]. A study conducted in Indonesia found that exposure to NO_2_ increases children’s risk of developing an acute respiratory infection [111]. This evidence is confirmed by other studies, such as the one conducted by Darrow and colleagues, who observed independent effects of O_3_ and primary traffic pollutants (NO_2_, CO, and the elemental carbon fraction of PM_2.5_) on hospital admissions for pneumonia and upper respiratory infections among children between 0 and 4 years old; moreover, the study highlights the carbon component in the PM_2.5_ mixture as particularly damaging for respiratory infections in children [82]. Exposure to PM has been shown to facilitate Streptococcus pneumoniae infections by improving its ability to adhere to lower airway cells and impair bacterial clearance. However, the role of low PM concentrations, as commonly found in high-income urban environments, on pneumococcal pneumonia is still unclear [112]. A study has shown that PM_10_ favors the replication of RNA viruses through the inhibition of innate immunity and the upregulation of several genes related to metabolic pathways, supporting the idea that PM_10_ exposure is associated with increased severity of pulmonary infections [113].

A Chinese study also shows that exposure to highly polluted areas increases the risk of tuberculosis, especially in children exposed to PM_10_, SO_2_, and NO_2_ [89].

Furthermore, several studies conducted in adults have linked long-term exposure to air pollutants to novel human coronavirus disease 2019 (COVID-19) infection and mortality rates [114,115]. In particular, a recent English study [116] found that an increase of 1 μg/m^3^ in PM_2.5_ and PM_10_ levels increases infection risk by 12% and 8%, respectively, and an increase of 1 μg/m^3^ in NO and NO_2_ levels spreads COVID-19-related mortality risk by 1.5% and 2.5%, respectively. Although the underlying mechanism is unclear, several hypotheses have been proposed, including the inflammatory state promoted by long-term exposure to pollutants and inhibition of the antimicrobial response. However, the relative weight of air pollution in increasing cases of SARS-CoV-2 infection has yet to be definitively determined, due to the several confounding factors and the numerous variables to take into consideration, such as particles’ size and chemical composition, the population’s age distribution, density, and social habits, the restrictive measures applied, and the weather conditions [117].

Because of their tiny size, large surface area, and high density, deposition, and penetration, UFPs may be more toxic than larger air pollutants. UFPs are associated with increased access to the emergency department for main respiratory diseases, such as pneumonia (RR 1.2; IQR 1.04–1.38), bronchitis (RR 1.17; IQR 1.03–1.33), and upper respiratory infections (RR 1.14; IQR 1.02–1.28) [74]. The relative risk of such exacerbations is positively correlated with UFP concentration [74].

### 4.4. Lifelong Effects on Respiratory Function

Injuries to the developing lung, both during pregnancy and in the first years of life, have been shown to have long-term effects on the respiratory system. In the long term, pollution can damage lung function, affecting patterns of growth and causing early function decline, as measured by forced expiratory volume in one second (FEV_1_) and forced vital capacity (FVC) [118]. A study by Gauderman et al. [119] followed 1759 patients over eight years and demonstrated that lung development (assessed by measuring FVC, FEV_1_, and maximum mid-expiratory flow, known as MMEF) was reduced in children exposed to higher levels of air pollution. These deficits have been associated with NO_2_, SO_2_, PM_2.5_, and EC, and the authors hypothesized that the main mechanisms involved were a reduction in alveolar growth and airway inflammation. Molter et al. [79] demonstrated that long-term exposure to PM_10_ and NO_2_ is associated with reductions in lung volume growth.

Similarly, pre- and post-natal PM_2.5_ exposure can decrease the number of alveoli in mice [120]. O_3_ exposure is associated with acute lung function reduction in healthy and asthmatic children, and toxicologic studies involving animals show that O_3_ negatively affects lung development. Similarly, acute and chronic NO_2_ exposure can impair lung function [16]. Long-term PM_2.5_ exposure is associated with slower lung function growth in children; on the contrary, the evidence for PM_10_ is less robust due to the lower availability of long-term models [80]. Some studies even failed to find an association between long-term PM_10_ exposure and child lung function [121,122].

The exposure during lung development seems to be associated with a negative impact on lung function that may ultimately result in COPD. Low childhood lung function is a risk factor for developing COPD or asthma-COPD overlap syndrome in adulthood, according to a study of 1389 patients who underwent spirometry in childhood and then in adulthood [90]. The main reasons why exposure to air pollutants in early life can lead to COPD in adults are (1) poor lung function; (2) recurrent respiratory illness [91,92]; (3) pro-inflammatory changes in the immune system (enhancement of Th2 and Th17 responses); and (4) dysregulated anti-viral immune responses [50]. Moreover, air pollution is responsible for short stature and stunting, which harm lung development [123].

## 5. Prenatal Exposure

The development of the human respiratory system is a complex process that begins early during the embryonic stage and lasts long after birth, with complete alveoli development occurring in adolescence. Growth and transcription factors and their interactions with the extracellular matrix coordinate all the steps in lung morphogenesis. During its differentiation and development, the lung is highly vulnerable to exposure to environmental pollutants [91]. Recent evidence has shown that children are harmed not only by direct exposure to pollutants but also by in-utero exposure, which can cause alterations in lung development (pollutants can affect cellular differentiation, morphogenesis, and vascularization) and increase the risk of respiratory illness from childhood into adulthood [15,61]. Most non-human evidence shows that prenatal exposure to PM_2.5_ and PM_0.1_ alters lung and immune system development, increasing the risk of acute and chronic pulmonary diseases (e.g., respiratory infections and asthma, respectively).

Epidemiological studies suggest maternal exposure to environmental hazards, such as combustion-generated PM, is associated with an increased childhood asthma incidence [86]. According to Wang and colleagues [108], pollution appears to cause systemic oxidative stress in pregnant mice and inhibit Th1 lung maturation, leading to asthma in the offspring. The retrospective cohort study by Zhang et al. [87] highlighted strong evidence that early-life exposure to submicrometric PM increases the risk of asthma among preschool children. All size fractions of PM can contribute to asthma, but it was especially notable for PM_1_, and the effect of prenatal exposure to PM_1_ was more pronounced in males [88]. The study performed by Lavigne et al. found that exposure to UFPs during the second trimester of pregnancy was associated with an increased risk of childhood asthma [85].

Air pollution, particularly TRAP, is recognized to be an important predisposing factor for childhood pneumonia, and the retrospective cohort study performed by Lu and colleagues highlighted that preconception (up to one year before conception) and prenatal exposure to industrial air pollution play a role in the onset of childhood pneumonia [124]. Similarly, a Spanish study reveals that exposure to NO_2_ during pregnancy is associated with an increased risk of respiratory infections in infants [73].

The study by Latzin et al. discovered that prenatal exposure to outdoor air pollution is associated with airway inflammation (NO_2_) and higher ventilatory demand (PM_10_) [81]. Furthermore, maternal exposure to ambient TRAP, especially NO_2_, PM_10_, and PM_2.5_, was also proven to harm children’s lung function and development [61,67]. In addition, prenatal exposure to air pollution is associated with reduced fetal lung growth [16,118] and lower levels of forced mid-expiratory flow (FEF_25–75_) and FEV_1_ [125,126].

Air pollution exposure during pregnancy and after birth can eventually alter the adult lung function trajectory, leading to multimorbidity [127].

Moreover, several meta-analyses show that prenatal exposure to air pollution increases the risk of intrauterine growth restriction and low-birth-weight offspring [128,129], mainly due to PM_2.5_, PM_10_, NO_2_, and O_3_ exposure both in the first and last month of pregnancy [130]. In addition, maternal outdoor air pollution exposure is also associated with the risk of very early and early preterm births [131], which can result in bronchopulmonary dysplasia.

The effects of prenatal exposure to air pollutants on respiratory disease are summarized in Table 2.

## 6. Concluding Remarks

Outdoor air pollutants affect children’s lung development, even during pregnancy, increase the risk of acute illnesses such as respiratory infections, bronchiolitis, and asthma, and eventually can contribute to COPD and decreased lung function in adulthood. Evidence suggests that there is no safe limit for air pollution exposure [132]. Different types of evidence support the importance of implementing air quality policies to prevent children’s respiratory morbidity; for instance, a study carried out in California in 2015 [133] highlighted that long-term improvements in air quality are associated with statistically and clinically significant positive effects on lung function growth in children. Other studies conducted during the SARS-CoV-2 pandemic lockdown showed that restrictions on motor vehicle circulation could reduce pollution and, consequently, acute respiratory disease [101,102].

Pollution is a more significant topic in low-income settings. Air pollution inequality widens not only between rich and developing nations but also between rich and poor people within the same country. The latter tend to live in polluted suburbs, with more highways and worse air quality. Similarly, lower-income countries tend to have lax regulations regarding air pollution and vehicle emissions [134,135].

While governments should enforce air quality standards by monitoring air pollution levels and adopting policies to reduce pollution, pediatricians could suggest other measures to mitigate exposure to traffic pollution, such as (1) using masks, (2) preferring active transport rather than motorized vehicles, (3) choosing travel routes that minimize vehicle air pollution exposure, (4) monitoring air pollution levels and consequently avoiding outdoor activities when and where air pollution is higher, and (5) preferring a healthy diet rich in antioxidants or anti-inflammatory agents [136].

Future national and international policies should prioritize public health issues, especially in poorer countries and suburbs, to enhance air quality, mitigate the harmful effects of pollutants, and ultimately improve children’s health.

Despite the considerable consistency of epidemiological findings irrespective of the specific physicochemical properties of the major PM components, in the last decade, more and more studies have suggested that pro-inflammatory and oxidative responses to the respiratory system in humans are driven by specific sources of pollution in combination with the general state and dynamics of the atmosphere (e.g., atmospheric mixing and stability, temperature, wind speed, precipitation, etc.) [137,138,139,140]. In this context, the toxicity of PM may derive predominantly from properties (e.g., the content of BC or other micropollutants and their mixing state, ROS species, UFPs, etc.), which do not necessarily correlate to PM total mass. For these reasons, air quality policies should focus on sources that emit components that mainly contribute to health hazards. This requires further investigation to link single components of air pollutants to specific health outcomes.

## Figures and Tables

**Figure 1 ijms-24-04345-f001:**
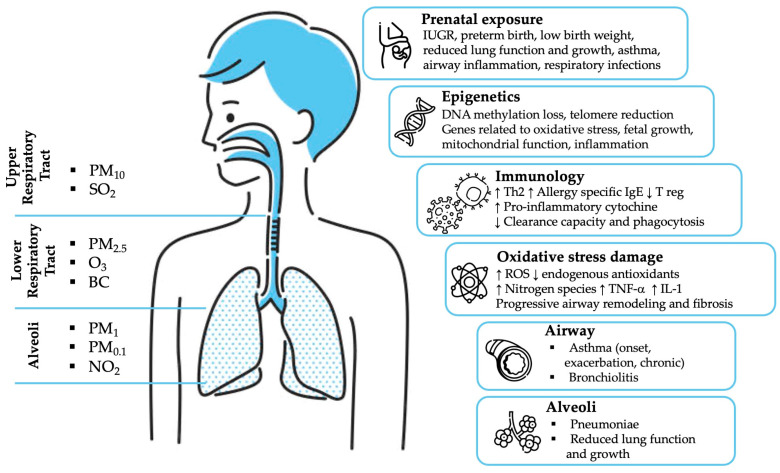
Regional deposition of major air pollutants and main direct and indirect mechanisms of damage to the respiratory system. PM_2.5_: Particulate Matter with an aerodynamic diameter of 2.5 µm; PM_10_: Particulate Matter with an aerodynamic diameter of 10 µm; PM_0.1_: Particulate Matter with an aerodynamic diameter of 0.1 μm; SO_2_: Sulphur Dioxide; O_3_: Ozone; BC: Black Carbon; NO_2_: Nitrogen Dioxide; IUGR: Intrauterine Growth Restriction; ROS: Reactive Oxygen Species; TNF-α: tumor necrosis factor; IL-1: Interleukin-1; ↑ increase; ↓ decrease.

**Figure 2 ijms-24-04345-f002:**
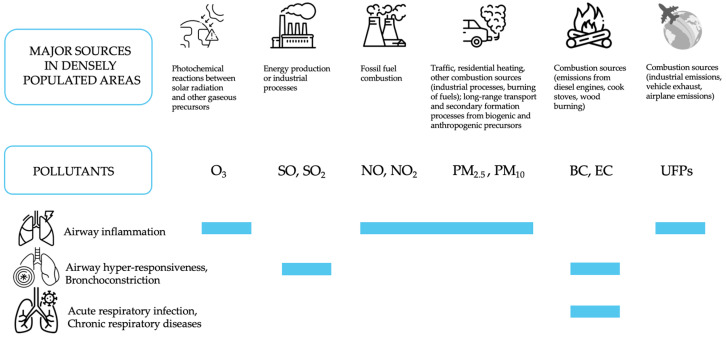
Main air pollutants, their major sources, and their effects on the respiratory system. O_3_: Ozone; SO, SO_2_: Sulphur oxides; NO, NO_2_: Nitrogen oxides; PM_2.5_: Particulate matter with an aerodynamic diameter of 2.5 µm; PM_10_: Particulate matter with an aerodynamic diameter of 10 µm; BC: Black Carbon, EC: Elemental Carbon; UFPs: Ultra Fine Particles.

**Figure 3 ijms-24-04345-f003:**
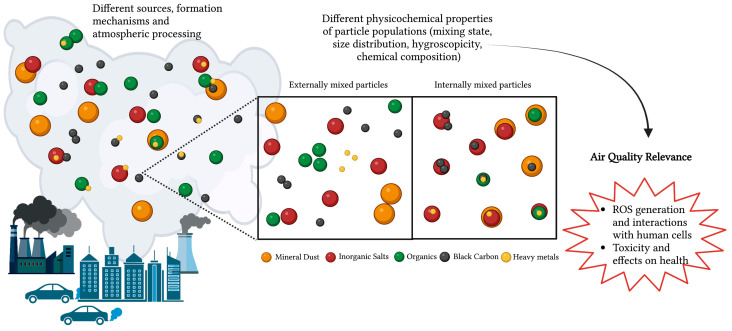
The figure exemplifies the concept of how different mixing structures of particle populations are characterized by specific physicochemical properties that determine, in terms of air quality relevance, how efficiently particles deposit along the human airway after inhalation and interact with human cells, thus driving the associated toxicity. The mixing state of particles can be conceptually classified as either internally mixed (whether all particles in the population contain the same species in the same mass fractions) or externally mixed (if every particle contains just one species or surrogate species, such as primary organics). Of course, in real-world conditions, the mixing state will be in between those two extremes. Most internally mixed particles are formed through atmospheric processing, leading to aged particles with different species, such as, for instance, inorganic salts (nitrates and sulfates) and organics, or also mineral dust, within the same particles. In addition, while fresh particles directly emitted from sources are more frequently externally mixed, some internally mixed particles are directly emitted from combustion sources (e.g., BC associated with freshly emitted organics and heavy metals). The terms external and internal mixture can also be applied to particle types within the particle population, such as particles in the same size range, particles from the same source, particles with similar composition, or particles with similar properties.

**Figure 4 ijms-24-04345-f004:**
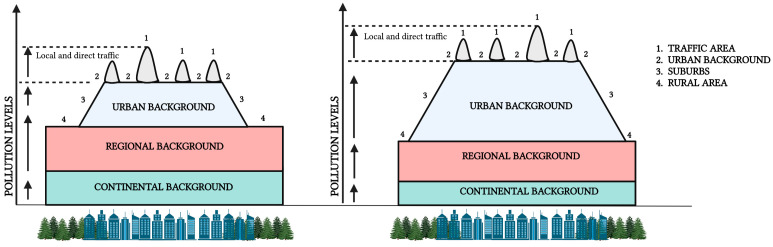
Typical patterns of traffic-related air pollution (TRAP) exposure across an urban environment. In this representation, we can identify areas in the proximity of major TRAP local sources characterized by enhanced concentration (marked by number 1) above the urban, regional, and continental backgrounds (marked by 2, 3, and 4, respectively). In general, air pollution at a specific point in an urban area can be partitioned into the amount associated with the regional background entering the same area, the urban background from dispersed primary source emissions, which include traffic, and processed secondary products of these emissions and other possible nearby sources. In addition, the figure depicts how the relative impact of those contributors varies spatially and temporally in relation to meteorological conditions and source profiles at different scales. This suggests that exposure to the same PM mass concentrations can be associated with different physicochemical properties (e.g., mixing state, size distribution, and chemical composition of the particle population, as described in the previous section) and substantially different health outcomes. This Figure has been adapted from [23,38].

**Figure 5 ijms-24-04345-f005:**
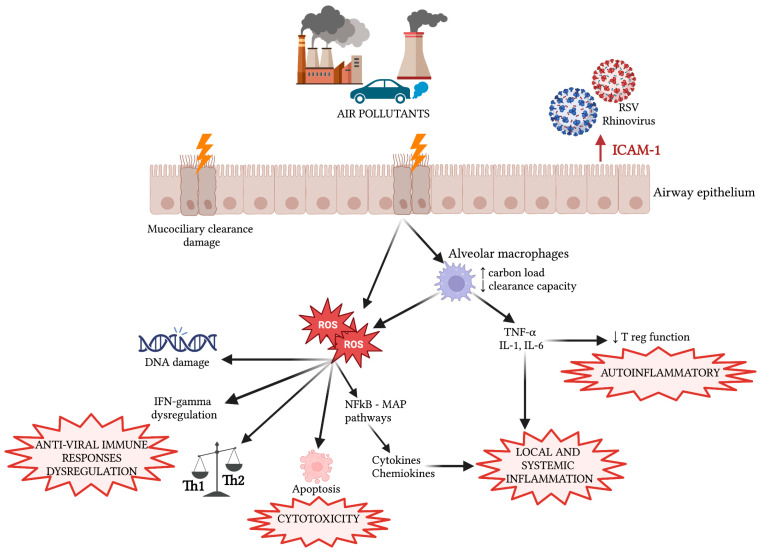
Main mechanisms of damage of air pollutants on the respiratory system. ROS: Reactive Oxygen Species; ICAM-1: Intercellular Adhesion Molecule-1; IFN: interferon; IL: Interleukin; TNF-α: Tumor Necrosis Factor Alfa.

**Table 1 ijms-24-04345-t001:** Effects of air pollution on the immune system. O_3_: Ozone; NO_2_: Nitrogen Dioxide; PM: Particulate Matter; PM_2.5_: Particulate Matter with an aerodynamic diameter of 2.5 µm; SO_2_: Sulphur Dioxide; ROS: Reactive Oxygen Species; GM-CSF: Granulocyte-Macrophage Colony-Stimulating Factor; ICAM-1: Intercellular Adhesion Molecule-1; IFN: interferon; IL: Interleukin; TNF-α: Tumor Necrosis Factor Alfa; ↑ increase; ↓ decrease.

Innate Immune System	Epithelial cells	↑ Pro-inflammatory cytokines: IL-1, IL-6, TNFα, GMCSF (O_3_, NO_2_, PM) ↑ ICAM-1 expression (NO_2_) ↑ Leukotriene C4 (NO_2_)
	Monocytes/Macrophages	↑ Carbon loading; ↓ Clearance capacity and phagocytosis; ↑ Pro-inflammatory cytokine (PM)
	Neutrophils	↑ Migration and activation (PM, O_3_, NO_2_)
	Eosinophils	↑ Migration and activation (SO_2_, PM_2.5_)
Adaptive Immune System	Immune tolerance	↓ T reg function; neo-antigens (ROS); ↑ antigens immunogenicity (PM); ↑ immunogenicity of antigens (PM)
	B cells	↓ IgA and ↑ IgE (PM_2.5_)
	T cells	↓ Th1 and ↑ Th2 (NO_2_, PM); ↑ Th17 response (PM); Dysregulated IFN-gamma and IL-17A production (PM)

**Table 2 ijms-24-04345-t002:** Outdoor air pollutant effects on respiratory disease and related issues: prenatal exposure, short- and long-term effects. TRAP: traffic-related air pollution; NO_2_: Nitrogen Dioxide; PM: Particulate Matter; PM_2.5_: Particulate Matter with an aerodynamic diameter of 2.5 µm; PM_10_: Particulate Matter with an aerodynamic diameter of 10 µm; UFPs ultrafine particles; BC: Black Carbon; O_3_: Ozone; SO_2_: Sulphur Dioxide; CO: carbon monoxide; COPD: Chronic obstructive pulmonary disease.

Prenatal Exposure	Short Term Exposure	Long Term Exposure
Reduced lung function and growth (TRAP, NO_2_, PM_10_, PM_2.5_) [61,67]	▪ Bronchiolitis (PM_10_ [68], PM_2.5_ [68], NO [68], NO_2_ [68], UFPs [69]) ▪ Severe Bronchiolitis (PM_2.5_ [70], PM_10_ [70], NO_2_ [18]) ▪ RSV Bronchiolitis (NO [68,71], NO_2_ [68,71], PM_2.5_ [68,71], PM_10_ [69,72]	Reduced lung function (NO_2_, O_3_) [16]
Increased risk of Respiratory Tract Infection (NO_2_) [73]	Asthma exacerbation (UFPs [74], O_3_ [75], SO_2_ [76], NO_2_ [77], TRAP [78], BC [78], PM_2.5_ [78])	Reduced lung growth (PM_2.5_, PM_10_, NO_2_) [79,80]
Airway inflammation (NO_2_) [81] Higher ventilatory demand (PM_10_) [81]	▪ Pneumonia (UFPs [74], NO_2_ [82], PM_10_ [82], PM_2.5_ [82], TRAP [82]), ▪ Severe pneumonia (O_3_, PM_2.5_) [83]	Chronic Asthma (PM_2.5_, TRAP) [78,84]
Increased incidence of asthma (UFPs [85], PM_0.1_, PM_1_, PM_2.5_) [86,87,88]	Tuberculosis (PM_10_, SO_2_, NO_2_) [89]	COPD [90,91,92]
Allergic rhinitis (TRAP [93], NO_2_ [94])	Upper Respiratory Tract Infection (UFPs [74], PM_2.5_ [95], PM_10_ [95], SO_2_ [95], NO_2_ [95], CO [95])	Allergic rhinitis (TRAP [93], PM_10_ [93], SO_2_ [94], NO_2_ [93])

## Data Availability

No new data were created or analyzed in this study. Data sharing is not applicable to this article.

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
