# Peer review of "Outdoor Air Pollution and Childhood Respiratory Disease: The Role of Oxidative Stress"

_ijms, 2023, doi:10.3390/ijms24054345_

Round 1

Reviewer 1 Report

I appreciate the opportunity to critically review this manuscript in which the authors summarize the relationship between environmental pollution and respiratory diseases in children. The manuscript is well-written and I find the information presented to be clear and comprehensive. I issue a couple of minor comments that I put for the consideration of the authors.

Although governments indeed must monitor PM10 and PM2.5 particles, this may be little attended to, particularly in countries with a low economic level. It would be convenient to quote it briefly.

Recently published data (i.e. DOI: 10.1016/j.envpol.2020.115859) suggest a relationship between levels of environmental contamination and SARS-COV-2 activity and mortality. It might be interesting to discuss it briefly.

Reviewer 2 Report

Comprehensive review that satisfactorily covers all the individual sections and documents the effects of environmental pollution on the respiratory system of children. It adequately documents the association of environmental pollution and respiratory diseases in childhood.

Reviewer 3 Report

Dear author,

After reviewing the article I have found that most of the data is already known. Most important part is representation which is poor in current form.

The article seems monotonous and not-interesting to read. Addition of some more images is required for pathways or effect that represent the literature more effectively than writing.

Make article information more critical. 

Good Luck

Round 2

Reviewer 3 Report

Dear authors.

The manuscript has been improved in comparison to previous versions but still some minor mistakes in using words or english terminologies are left.

Good luck.
